✿ PLOS | ONE

# Exploring the role of the various methionine residues in the *Escherichia coli* CusB adapter protein

Aviv Meir[1], Gulshan Walke[1], Fabian Schwerdtfeger[1,2], Lada Gevorkyan Airapetov[1], Sharon Ruthstein[1]*

1 Department of Chemistry; Bar-Ilan University, Ramat Gan, Israel, 2 Department of Chemistry and Pharmacy, Albert-Ludwigs-University, Freiburg, Germany

* Sharon.ruthstein@biu.ac.il

**Data Availability Statement:** All relevant data are within the paper and its Supporting Information files.

## Abstract

The dissemination of resistant pathogenic microbes has become one of the most challenging problems that modern medicine has faced. Developing novel drugs based on new molecular targets that previously were not targeted, is therefore the highest priority in antibiotics research. One approach that has been recently suggested is to inhibit copper transporters in prokaryotic systems. Copper is required for many biological pathways, but sometimes it can harm the cell. Pathogenic systems have a highly sophisticated copper-regulation network; therefore, a better understanding of how this network operates at the molecular level should assist in developing the next generation of antibiotics. The CusB protein is part of the CusCBA periplasmic Cu(I) efflux system in Gram-negative bacteria, and was recently reported to play a key role in the functioning of the whole CusCBA system, in which conformational changes as well as the assembly/disassembly process control the opening of the transporter. More knowledge of the underlying mechanism is needed to attain a full understanding of CusB functioning, which is associated with targeting specific and crucial residues in CusB. Here, we combine in-vitro structural measurements, which use EPR spectroscopy and UV-Vis measurements, with cell experiments to explore the role of the various methionine residues in CusB. We targeted two methionine residues (M227 and M241) that are essential for the proper functioning of CusB.

## Introduction

Copper is an essential nutrient for aerobic organisms; its ability to exchange electrons as it cycles between cuprous and cupric states has been harnessed by enzymes that catalyze a wide variety of biochemical processes [1, 2]. However, this redox activity also confers copper with toxic properties when it is present in the free ionic form: free copper can participate in Fenton-like chemical reactions to generate the highly toxic hydroxyl radical from hydrogen peroxide and superoxide [3–5]. Owing to its toxicity, copper has been known as an antimicrobial agent for thousands of years [6]. It is therefore not surprising that microbes have developed

**Funding:** SR acknowledges the support of ISF grant no. 176/16.

**Competing interests:** The authors have declared that no competing interests exist.

**Abbreviations:** CusCBA, Heavy metal efflux complex; CueO, Copper oxidize; CopA, Copper transporter; CueR, Copper efflux regulator; CusB_NT, CusB N-terminal domain; UV-Vis, Ultraviolet Visible spectroscopy; EPR, Electron paramagnetic resonance; DEER, Double electron electron resonance; SDSL, Site directed spin labeling; PDB, Protein data bank.

tightly regulated mechanisms for copper transport and intracellular distribution, to maintain negligible changes (subfemtomolar concentrations) in their intracellular levels [7–9]. Selective inhibition of the copper efflux in microbes is expected to raise their copper levels, accelerate the Fenton reaction, and augment the production of free radicals. Ultimately, this chain of events kills the microbes.

Four different mechanisms regulate cellular Cu(I) in prokaryotic systems [10]: (i) the cytoplasmic CueR- metal sensor initiates the transcription process of CueO and CopA associated with Cu(I) coordination [11]; (ii) CopA uses the energy provided by ATP hydrolysis to drive the efflux of Cu(I) from the cytoplasm to the periplasm [12]; (iii) CusF, a metallochaperone, transfers Cu(I) from CopA to the CusCBA efflux system, which pumps Cu(I) from the periplasm to the extracellular region [13]; and (iv) CueO oxidizes Cu(I) to its less toxic oxidation state, Cu(II) [14]. Although homologs of CopA and CueO also exist in the human cells ATP7b and SOD, respectively, the CusCFBA and CueR systems are unique to prokaryotic systems. Thus, targeting the latter two copper transport systems with antibiotics might be very beneficial in terms of freedom from an antibiotic effect on human copper-transport systems. CueR is a metal sensor that senses the Cu(I) ion with high affinity and induces a transcription process [11, 15]. Metal sensors such as CueR exist in all microbes, but their structure and functionality vary among microbe types. In contrast, Cus efflux systems are found in many pathogenic microbes such as Legionella, Salmonella, Klebsiella, Pseudomonas, and many other Gram-negative bacteria. The sequence identity among all species is about 30%, and their structures and functions should be similar in all microbes.

The Cus complex comprises an inner membrane proton-substrate carrier (CusA) and an outer membrane protein (CusC). The inner and the outer membrane proteins are connected by a linker protein, CusB, in the periplasm, which is composed of CusA:CusB:CusC in a oligomerization ratio of 3:6:3 [16]. Owing to the large size of the system, the structure of the entire complex has not been elucidated; however, the crystal structures of the individual components (CusC, CusA, and CusB) and the CusBA complex have been determined [17–19]. Padilla-Benavides *et al.* [20] have successfully demonstrated a specific interaction between the CopA transporter extracellular domain and CusF. Based on X-ray absorption spectroscopy, Chacón *et al.* [21] suggested that the CusB-CusF interaction functions as a switch for the entire Cus efflux system and facilitates the transfer of copper to the CusA component. Taken together, these findings indicate that in the periplasm, Cu(I) is transferred from CopA to CusF and then to the CusCBA complex through direct and specific interaction (Fig 1A). The crystal structure of *E. coli* CusB indicates that the protein is folded into an elongated narrow structure that comprises four domains with a flexible hinge between domains 2 and 3 of the protein [18]. Recently, we used electron paramagnetic resonance (EPR) spectroscopy-based nanometer distance measurements to show that CusB undergoes major structural changes associated with Cu(I) binding [22]; this results in two structures for the CusB dimer in solution: apo-CusB and holo-CusB (Fig 1B). The EPR data suggests a more compact structure upon Cu(I) binding, in agreement with the gel filtration chromatography experiments [23]. Moreover, Chen *et al.* showed that under copper stress, CusB changes its conformation and shifts the equilibrium from a disassembled CusCBA complex to an assembled one. Methionine residues in the CusB play a critical part in Cu(I) binding [24]. Bagai *et al.* revealed that four out of ten methionine residues (M49, M64, M66, and M311) in CusB are conserved [23]. Moreover, the crystal structure suggests that there are two methionine residues ((i) M190, domain 4, and (ii) M324, domain 1), which are not conserved, yet take part in Cu(I) binding [18]. The most studied region in CusB is the N-terminal domain (CusB_NT), which comprises 60 amino acids (residues 28–88) that were shown to interact directly with CusF [25–27]. Studies on CusB knockout (ΔCusB) and CusBΔNT have revealed a Cu(I)-sensitive phenotype in cells and indicated that

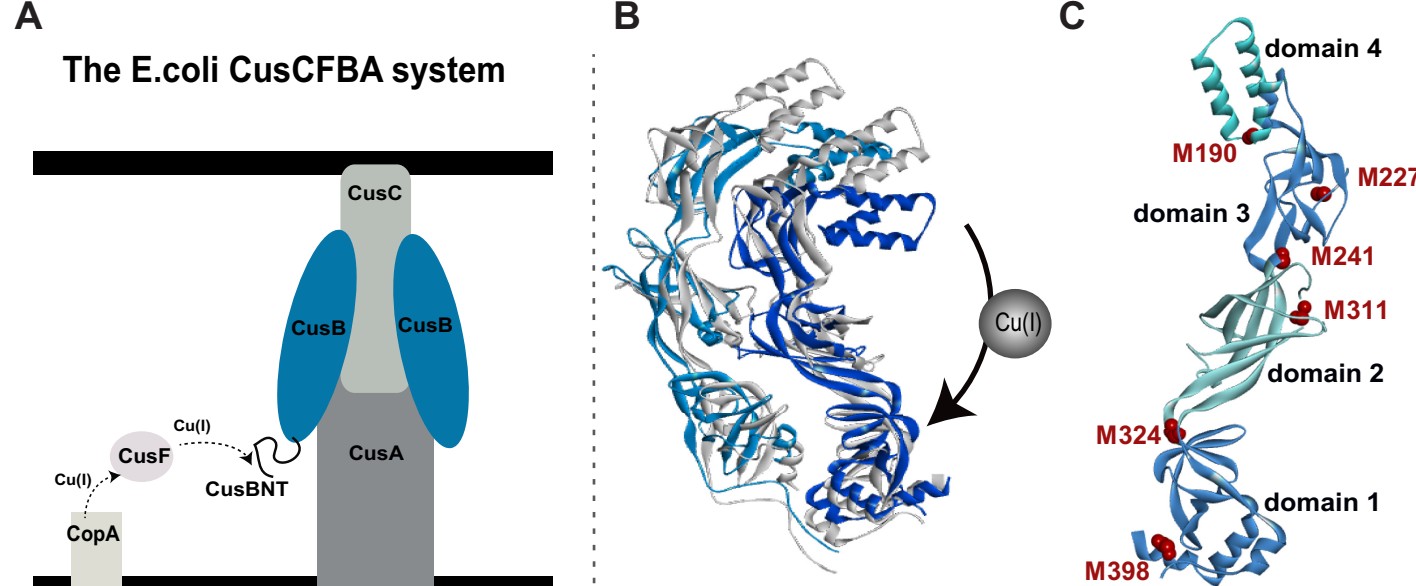

**Fig 1. The CusCBA efflux system.** A. A schematic view of the CusCFBA periplasmatic efflux system. Cu(I) is transferred from the CusF metallochaperone to the N-terminal domain of CusB (CusB_NT) and out of the cell through CusB and CusC proteins by an unknown transfer mechanism. B. The CusB dimer structural models for apo-CusB (gray protein) and holo-CusB (blue colors) based on EPR constraints. C. The positions of the methionine residues (red colors) in the crystal structure of CusB monomer (PDB 3H94); residues 1–88 are missing from the crystal structure.

M64 and M66 of CusB_NT are important residues, both for Cu(I) coordination to CusB_NT as well as for its interaction with CusF, whereas M49 of CusB_NT is important for interacting with CusF [21, 26–28]. Coupled with various constructs of ΔCusB, the McEvoy group found that CusB_NT is not fully functional by itself without the remaining part of the protein, and ruled out the idea that CusB acts only as a metal chelator [26]. They found that it is also involved in conformational changes resulting in a more compact form [23]. To date, Cu(I) binding sites in domain 2 and domain 3 of the CusB have not been reported. Since this region was found to undergo structural changes upon Cu(I) binding, we suspect that there may be additional Cu(I) sites that have not been targeted yet.

In the present study, we use a combination of in-vitro spectroscopic measurements and cell experiments to study the significance of each single methionine residue in CusB and its effect on cell survival. We demonstrate that two additional methionine residues of CusB, namely M227 (Domain 3), and M241 (Domain 2) (Fig 1C), are essential for cell growth and viability. Pulsed EPR spectroscopy confirms that the conformational changes in CusB, which were associated with Cu(I) binding, depend on these Met residues. This study emphasizes the role of domains 2 and 3 of CusB in the functioning of CusCBA transporter and regulation of in-cell copper concentration.

## Results and discussion

### Copper stress in CusB mutants during cell growth

The present study aimed to elucidate the various Cu(I) binding sites in CusB; we hypothesized that deletion of single methionine (Met) residues in CusB, which possibly play a role either in Cu(I) coordination [18, 25–27] in the Cus channel assembly process, or in the transfer mechanism, may also affect cell viability upon Cu(I) stress. Ten Met residues exist in CusB; in order to determine which of them are essential for cell growth, each time one methionine residue

was mutated to isoleucine. Initially, we knocked out CusB (ΔCusB) in the native *E. coli* cells and then transformed either WT or a mutated gene into the knocked out cells. The protein expression level was identical for all clones; therefore, we could compare the growth rates of the various mutants (S1 Fig). We used Cu(II) in the cell experiments because it can penetrate into the cell and is reduced to Cu(I) (S2 and S3 Figs). However, our conditions are not fully anaerobic; therefore, we also repeated the experiments in the presence of Cu(I), and the results were found to be similar (S4 and S5 Figs). This indicates that Cu(II) was reduced to Cu(I). The various clones were grown in M9 medium for 16 hr and their cell growth was compared in the absence and presence of Cu(II) in the medium. Growth rate values after 14 hr of cell growth for various CusB clones are presented in Fig 2.

The ability of wild-type (WT) CusB to grow under Cu(II) stress was found to be 3 ± 0.4% less than in the absence of Cu(II) after 14 hr of growth (Fig 2). We also monitored the cell growth of each single mutation (Fig 2). All mutants grew less well than WT-CusB even before adding Cu(II) to the growth solution, due to the slight effect of the minimal copper concentration naturally present in the M9 bacterial growth medium (0.5 μM), and the significance of the methionine residues in CusB for preserving copper homeostasis. However, the cell growth experiments succeeded in targeting three Met residues that mostly affect the *E. coli* cell growth after 14 hr under Cu(II) stress: CusB_M64I –a reduction of 30 ± 3%; CusB_M241I –a reduction of 25 ± 4%, and CusB M227I –a reduction of 18 ± 3%. CusB_M398I resulted in a reduction of 7 ± 2% after 14 hr with 3 μM Cu(II), which was the largest reduction among the remaining clones (Fig 2). All other mutants affected cell growth by less than 5%. M11I shows the lowest effect on the growth rate as expected, since it is located in CusB signal (residues 1–28) peptide which is cleaved upon maturation, suggesting that M11I does not affect this cleavage process. Interestingly, in comparison with previous results, smaller reductions in growth rate were observed for the conserved methionine residues and for the M190 and M324 residues, which were suggested to play a role in Cu(I) binding by crystallography [18, 25, 26]. In addition, previous in-vitro ITC measurements were not able to detect the functionality of M227 and M241 residues [23]. Based on the cell experiments, we decided to focus on the four

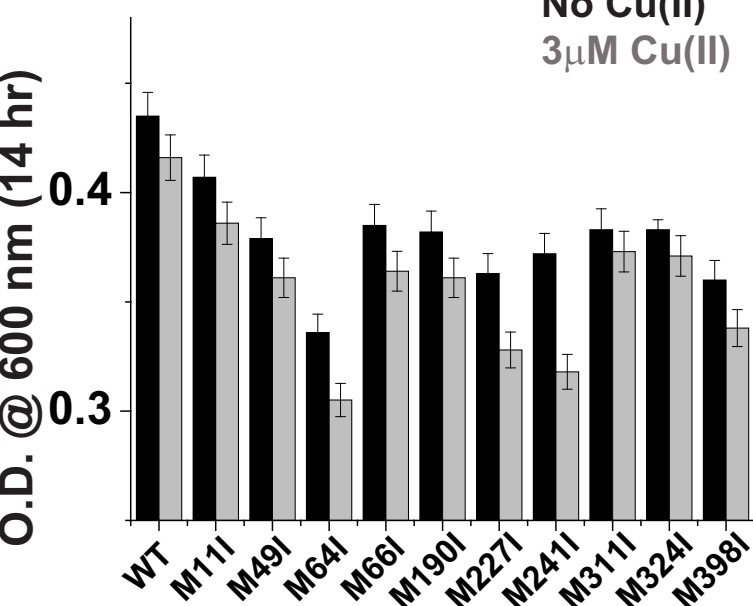

**Fig 2.** Cell growth rates for various CusB clones in the absence (black) and presence (gray) of 3 μM Cu(II) after 14 hr.

methionine residues which showed the largest effect on the growth rate, and may therefore form additional Cu(I) sites that were not identified earlier.

## Targeting conformational changes in CusB mutants by DEER measurements

Over the last decade, double electron-electron resonance (DEER) spectroscopy, coupled with site-directed spin labeling (SDSL), has been found to be an excellent means of obtaining structural and dynamic information on complex systems [29–36]. Hubbell *et al.* were the first to introduce the SDSL methodology in which they attached nitroxide spin labels to cysteine residues at selected positions within the protein [37, 38]; the most commonly used nitroxide spin label is the 1-oxyl-2,2,5,5-tetramethylpyrroline-3-methyl methanethiosulfonate spin label (MTSSL). The DEER technique, also called pulsed electron double resonance (PELDOR), is a pulsed electron paramagnetic resonance (EPR) technique used to measure dipolar interactions between two or more electron spins. Thus, it can provide nanometer-scale interspin distance information in the range of 1.5–8.0 nm. By using DEER and SDSL, we recently succeeded in targeting conformational changes in WT-CusB associated with Cu(I) binding [22]. We showed that CusB assumed a more compact structure (Fig 1B) upon Cu(I) binding, whereas most of the structural changes occurred in CusB domains 2 and 3. In order to explore the effect of methionine mutations on CusB conformation, we spin-labeled the protein at two sites: one at A236C (domain 3) and the second at A248C (domain 2). The DEER time domain signals are presented in Fig 3A and the corresponding distance distribution functions are in Fig 3B. Because CusB was found to be a dimer in solution [22], the distance distribution consists of all

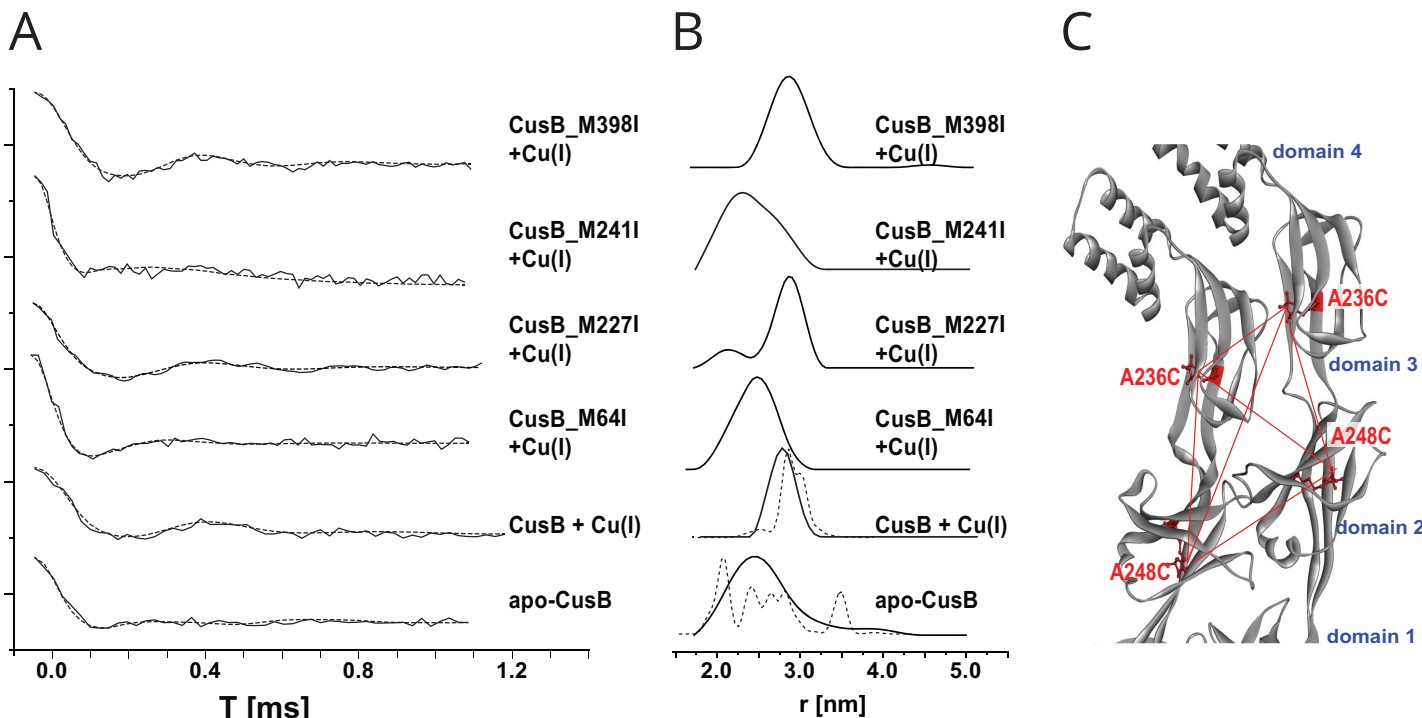

**Fig 3. Distance distribution measurements (DEER) on CusB mutants.** A. DEER time domain signals (solid lines) and fits (dashed lines) based on Tikhonov regularization for apo-CusB and CusB + Cu(I) (at a Cu(I):CusB ratio of 3:1) and various CusB + Cu(I) mutants spin labeled at the A236C and A248C positions. B. Corresponding distance distribution functions. The dashed lines represent calculated distribution functions obtained for the apo-CusB and CusB + Cu(I) structure models. C. The positions of the spin labels in WT-CusB (A236C,A248C). The red dotted lines denote the six distances targeted here under the broad distance distribution.

distances derived from the four spin-labels attached to the CusB dimer (Fig 3C). For WT-CusB, distance distributions of 2.5 ± 0.5 nm for the apo-state and 2.8 ± 0.3 nm for the holo-state (CusB+Cu(I)) were detected (Fig 3B). To compare the DEER data with the structure of CusB, we used the MMM software (2015 version) [39]. The software requires an entry PDB file, and the residues where the spin-labels are attached to the protein. This method describes spin labels by a set of alternative conformations or rotamers, which can be attached without serious clashes with atoms of other residues or cofactors. The rotamer library is derived from molecular dynamic simulation with a total length of 100 ns at a temperature of 175K, which is an estimate of the glass transition of a protein sample. Here we used the two structures of apo-CusB and holo-CusB, which were previously constructed based on various DEER constraints (Fig 1B) as an input PDB files for the MMM software [22]. The dashed black-line distribution in Fig 3B denotes the distance distribution obtained from MMM. The apo-state exhibits a broad distance distribution that is consistent with the calculated model structure of CusB, whereas in the holo-state the structure is more compact and rigid, which results in a much narrower distance distribution [22]. The orientation of the spin-labels in the two structures are presented in S6 Fig. DEER also confirmed that when one specific methionine residue among M64, M227, and M241 was mutated, structural changes in CusB protein associated with Cu(I) binding were affected. For simplicity, we used a linear composition of the apo-WT-CusB distance distribution with the holo-WT-CusB distance distribution (S6 Fig). Although the effect of each mutation on conformational changes may not be a simple two-step model between the apo- and holo-states, this approach can still provide us with an overview on how much each mutation disturbs the conformation and function of the CusB protein. In the case of CusB_M64I + Cu(I), the distance distribution is very similar to the apo WT-CusB distance distribution, suggesting that Cu(I) did not affect the conformation of CusB in the presence of M64I mutation. In contrast, for holo CusB_M398I, the distribution is slightly broader than the holo WT-CusB. However, the best fit was still obtained with 100% holo WT-CusB distance distribution, indicating that M398I did not affect the conformational changes of CusB upon Cu(I) binding. For CusB_M227I and CusB_M241I, the best fit was obtained with convolutions of 50% and 90%, respectively, of the apo-WT-CusB distance distribution with the holo-WT-CusB distance distribution. This suggests that both mutations affect the conformational changes of CusB in the presence of Cu(I), while M241I has a larger effect on conformation of CusB than the M227I mutation. The DEER agrees well with the cell experiments. When M398I, which hardly affects the cell growth, was mutated, the conformational changes induced by the Cu(I) binding was almost not affected. In contrast, M64I largely affected the cell growth consistent with the DEER data that showed that CusB did not assume any conformational changes upon Cu(I) binding. This suggests that Cu(I) needs to first bind to the N-terminal domain of CusB and then to transfer to the next site. Mutations in M227 and M241 which showed partial effects on cell growth, disturb the conformational changes induced by CusB in the presence of Cu(I), but do not completely inhibit it as does the M64I mutation. The DEER data indicate that mutations of M64, M227, and M241 affect both the cell growth and conformational changes in CusB and that the conformational changes presented in Fig 1B are essential for the protein to function.

## Cu(I) affinity to CusB

The affinity of CusB proteins for Cu(I) binding was evaluated by using the spectrophotometric competition assay with bicinchoninic acid (BCA) as a ligand (Eq 1).

$$P + ML_2 \rightarrow MP + 2L \tag{1}$$

where P corresponds to CusB protein, M corresponds to Cu(I), and L corresponds to BCA.

Cu(I) ions binding BCA with a stoichiometric ratio of 1:2. Cu(I)-(BCA)$_2$ show a characteristic high absorption at 562 nm. The binding affinity of proteins for Cu(I) can be calculated by monitoring the change in the absorption peak intensity of the Cu(I)-(BCA)$_2$ complex at 562 nm, which is associated with protein titration [40, 41].

When the solution of the Cu(I)-(BCA)$_2$ complex was titrated with CusB mutants, the CusB mutant competed for Cu(I) with BCA, thereby decreasing the concentration of Cu(I)-(BCA)$_2$, which lowered the absorption peak intensity at 562 nm. Fig 4A and 4B show the absorption spectra recorded for the titration of Cu(I)-(BCA)$_2$ with WT-CusB and CusB_M64I, respectively. The decrease in the absorption peak at 562 nm suggests that both WT-CusB and CusB_M64I may compete for Cu(I) with BCA. However, the affinity of CusB_M64I for Cu(I) is weaker than that of WT-CusB. Fig 4C presents changes in the absorption peak at 562 nm for various CusB mutants, as affected by Cu(I) concentration; it shows that WT-CusB binds Cu(I) with the highest affinity, whereas CusB_M64I has the lowest affinity for Cu(I).

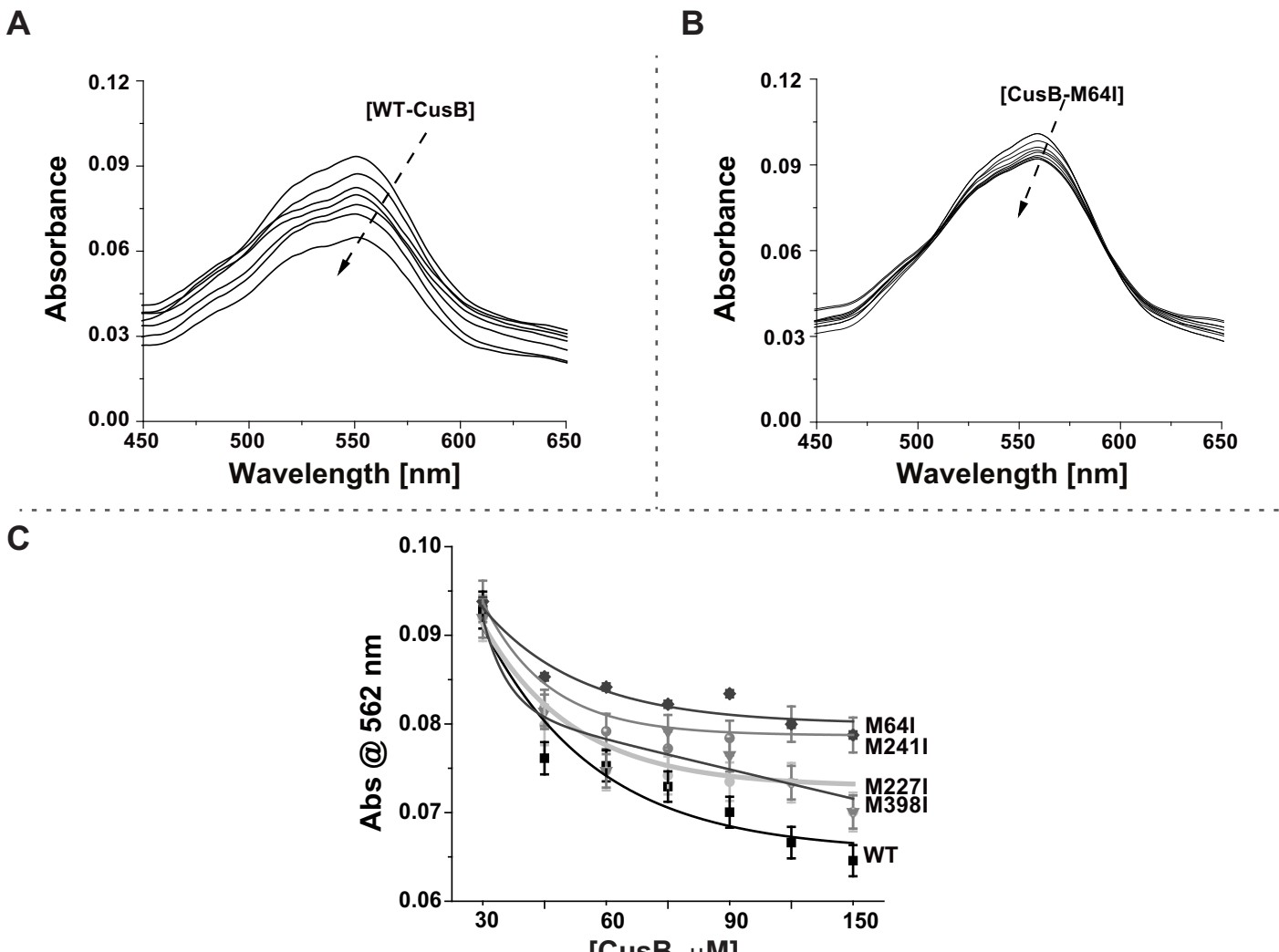

**Fig 4. Cu(I) affinity to CusB mutants.** Absorption spectra for Cu(I)-(BCA)$_2$ titrated with A. WT-CusB, B.CusB_M64I, C. Absorbance at 562 nm of Cu(I)-(BCA)$_2$ titrated with various CusB mutants.

The $K_D$ value for CusB-Cu(I) complexes was calculated according to a procedure developed by Xiao *et al.* [42]:

$$K_D\beta_2 = \frac{\left(\frac{[\text{CusB}]_{\text{total}}}{[\text{CusB}-\text{Cu(I)}]}\right) - 1}{\left\{\left([\text{BCA}]_{\text{total}}\big/\left[\text{BCA}_2 - \text{Cu(I)}\right]\right) - 2\right\}^2 \left[\text{BCA}_2 - \text{Cu(I)}\right]}, \tag{2}$$

where $\beta_2$ is a constant for [BCA$_2$-Cu(I)] formation [43], [BCA]$_{\text{total}}$ and [CusB]$_{\text{total}}$ are the concentrations used in the experiment, and [CusB-Cu(I)] and [BCA$_2$-Cu(I)] were calculated based on the absorbance changes that were normalized to the values without CusB (Table A in S1 File):

$$[CuB - Cu(I)] = [Cu(I)]\big/\left(1 - \frac{abs_{562}}{abs_{562,0}}\right) \tag{3}$$

$$[BCA_2 - Cu(I)] = [Cu(I)]\left(\frac{abs_{562}}{abs_{562,0}}\right) \tag{4}$$

To derive the $K_D$ values, we used $\beta_2 = 10^{17.2}$ M$^{-2}$ [40, 41]. The calculated $K_D$ values are listed in Table 1.

An increase in the $K_D$ value indicates that the Cu(I) binding to CusB is weaker. The lowest $K_D$ value was observed for WT-CusB; the highest value was for CusB_M64I; however, CusB_M241I had a slightly higher $K_D$ value than that of CusB_M227I. This result is very consistent with the DEER measurement, which showed more significant inhibition of the conformational changes in CusB associated with Cu(I) binding for CusB_M241I than for CusB_M227I.

## Cell viability decrease due to Cu(II) stress on CusB mutants during growth

We conducted live/dead fluorescence cell imaging experiments to determine whether point mutations may lead to cell death or only to a reduction in *E. coli* cell growth (Fig 5A, in which red and green dots denote dead and live cells, respectively). At the late lag phase, some dead cells were observed in the M9 medium, even for the WT-CusB clone, because M9 was a poor medium. Note that the late lag phase is different for all mutants; it is determined by the growth rate (S3 Fig). A comparison of cell images of the WT clone with those of the CusB_M64I clone at the late lag phase (Fig 5B) indicates a 62 ± 7% increase in the dead cell count before incorporating Cu(II) into the growth medium and a 100 ± 1% increase for clones grown under Cu(II) stress. With CusB_M241I and CusB_M227I, there were increases of 30 ± 3 and 23 ± 4%, respectively, in the dead cell count before incorporating Cu(II) into the growth medium and increases of 90 ± 2% and 75 ± 1%, respectively, for the same clones grown under Cu(II) stress.

**Table 1. Calculated $K_D$ values for CusB mutants.**

|  | $K_D$ [M];$\times 10^{-14}$ |
| --- | --- |
| **WT-CusB** | 8.7 ± 0.3 |
| **CusB_M64I** | 27.8 ± 0.4 |
| **CusB_M227I** | 15.6 ± 1.5 |
| **CusB_M241I** | 19.1 ± 0.4 |
| **CusB_M398I** | 8.9 ± 0.1 |

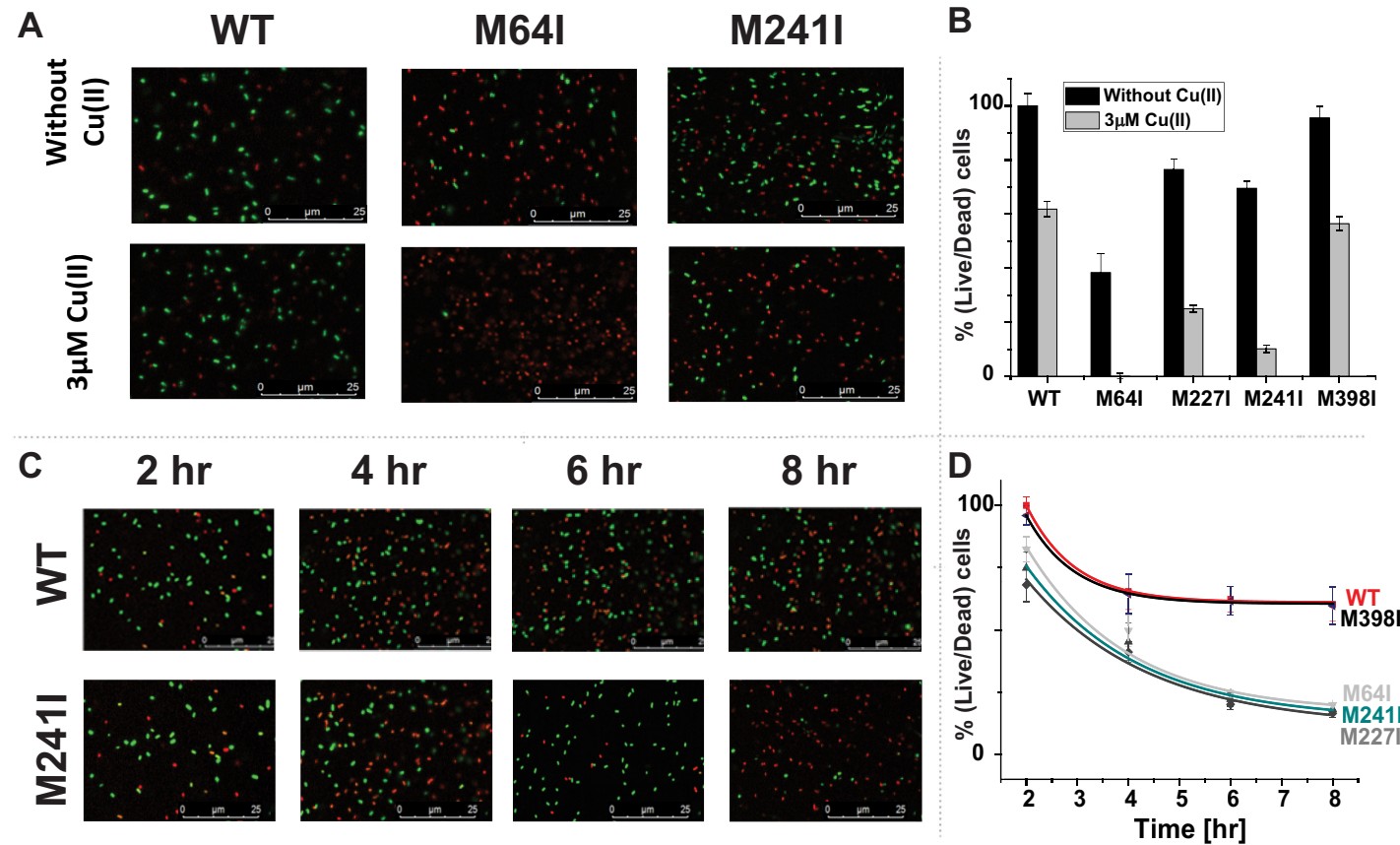

**Fig 5. Cell viability imaging.** A. Cell viability imaging for WT-CusB, CusB_M64I, and CusB_M241I; cells were grown with and without Cu(II) at the late lag phase (3–4 hr). Green dots denote live cells; red dots denote dead cells. B. Cell viability comparison at the late lag phase for all CusB clones in the presence and absence of Cu(II). C. Cell viability imaging for WT-CusB and M241I grown with Cu(II) as a function of time. Green dots denote live cells; red dots denote dead cells. D. Cell viability comparison for all CusB clones in the presence of Cu(II) as a function of time.

Monitoring the cell viability, as a function of time, revealed that changes in growth rates were followed by cell deaths and not merely slower cell growth (Fig 5C and 5D). After 8 hr of growth in 3 μM Cu(II), 60 ± 7% of cells containing WT-CusB remained alive and 18 ± 2% remained alive for CusB_M64I. The percentages of living cells after 8 hr of growth in 3 μM Cu(II) were 16 ± 2% and 20 ± 1% for CusB_M241I and CusB_M227I, respectively.

## Discussion

The CusCBA efflux system plays an indispensable role in copper homeostasis and cell viability within *E. coli* cells, and CusB controls the opening of the whole CusCBA complex. Understanding how CusB functions at the molecular level could facilitate the design of an inhibitor for the whole CusCBA efflux system. The trigger for opening the channel was shown to be the interaction between the metallochaperone CusF and the N-terminal domain of CusB (CusB_NT) associated with Cu(I) binding [25–27, 44]. Previously, we and others studied CusB_NT and elucidated the significance of the three conserved methionine residues (M49, M64, and M66) for Cu(I) binding within CusB_NT [21, 23, 26, 27]. In another study on full-length CusB, we found that CusB dimer had to adopt a specific conformation in association with Cu(I) binding [22]. Importantly, we showed that domains 2 and 3 underwent major structural changes associated with Cu(I) binding, in agreement with the crystal structure

reported; this suggests that an additional Cu(I) site exists in this region. In the present study, we initially performed cell growth experiments. Each time, we mutated a different methionine residue. All mutants affected the cell growth rate. However, four of them had a slightly larger effect on the cell growth compared to the others, and therefore we focused on these mutants. By coupling in-vitro and cell experiments, we highlighted the importance of three methionine residues: M64 (N-terminal domain), M227 (domain 3), and M241 (domain 2) for the growth rate and viability of the *E. coli* cells, and for the proper conformational changes in CusB. Pulsed EPR spectroscopy (DEER) showed that when one of the three methionine residues was replaced, it affected the conformational changes of CusB in the presence of Cu(I). Evaluating the $K_D$ value using UV-vis spectroscopy has shown that CusB binds Cu(I) ions with diminished affinity when it lacks one of the Met residues. Live/dead fluorescence cell imaging assisted in confirming that a mutation in one of the CusB methionine residues largely affects the cell viability. Time-dependent live/dead cell fluorescence imaging confirmed that CusB mutations lead to cell death and not merely to decreased cell growth. This finding indicates that introducing mutations in CusB results in toxicity that eventually leads to cell death within 8 hr. Our experimental data clearly indicates that at least one Cu(I) binding site exists in domain 2 and 3 of CusB. However, further study is required in order to better resolve this issue. Combining cell experiments with in-vitro experiments provided us with a deeper insight into the mechanism underlying the efflux system and an enhanced understanding of why point mutations decrease cell viability.

## Conclusions

By combining in-vitro structural measurements with cell experiments, we obtained better understanding of the mechanism of action underlying the adapter protein within the copper efflux system in Gram-negative bacteria, at the molecular and nanoscale levels. Our experiments in both solution and cells suggest that besides the essential methionine residues reported before, there are two additional methionine residues at CusB domains 2 and 3 (M227 and M241) that are important for proper function of the CusCBA transporter. Mutating these methionine residues affects the conformational changes of CusB in the presence of Cu(I) and reduces the functionality of the CusB protein, which decreases copper resistance and increases cell toxicity in pathogenic cells.

## Materials and methods

Experimental details and control experiments are described in the Supporting Information.

## Supporting information

**S1 Fig. Western blot gel for WT-CusB, ΔCusB, and CusB various mutants indicating on identical expression levels.**
(TIF)

**S2 Fig. Cell growth experiment.** OD after 16 hr for recombinant CusB, endogenic CusB, ΔCusB and BL21 cells with empty PYTB12 plasmid.
(TIF)

**S3 Fig.** Cell growth rates for various CusB clones in the absence (A) and presence (B), respectively, of 3 μM Cu(II).
(TIF)

**S4 Fig.** A. OD after 16 hr for native *E. coli* cells as affected by Cu(I) concentration. B. OD after 16 hr for ΔCusB cells affected by Cu(I) concentration. For ΔCusB cells, cell growth was terminated at a lower Cu(II) concentration. C. OD after 16 hr for ΔCusB-mutant cells as affected by Cu(I) concentration.
(TIF)

**S5 Fig. Effects of rich medium (LB broth) on cell growth.** A. OD after 16 hr for native E. coli cells grown in LB medium. B. OD after 16 hr for ΔCusB cells grown in LB medium as affected by 3mM Cu(II). C. OD after 16 hr for ΔCusB cells grown in LB medium as affected by 3mM Cu(I).
(TIF)

**S6 Fig.** A. Spin-label positions and orientations attached to CusB_A236C_A248C in apo and holo states. B. The distance distribution obtained and shown in Fig 3B. The red dotted lines corresponds to the linear composition a*f(r)+(1-a)*g(r); where f(r) is apo CusB distance distribution and g(r) is CusB+Cu(I) distance distribution.
(TIF)

**S1 File. Materials and methods.**
(PDF)

## Acknowledgments

SR acknowledges the support of ISF grant no. 176/16.

## Author Contributions

**Conceptualization:** Aviv Meir, Fabian Schwerdtfeger, Lada Gevorkyan Airapetov, Sharon Ruthstein.

**Data curation:** Aviv Meir, Fabian Schwerdtfeger, Lada Gevorkyan Airapetov, Sharon Ruthstein.

**Formal analysis:** Aviv Meir, Gulshan Walke, Lada Gevorkyan Airapetov.

**Investigation:** Aviv Meir, Gulshan Walke.

**Methodology:** Aviv Meir, Fabian Schwerdtfeger, Sharon Ruthstein.

**Project administration:** Sharon Ruthstein.

**Resources:** Sharon Ruthstein.

**Supervision:** Sharon Ruthstein.

**Writing – review & editing:** Aviv Meir, Sharon Ruthstein.

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
