## [Decision Letter · Decision Letter 0]

7 Aug 2019

PONE-D-19-17365

Exploring the role of the various methionine residues in the Escherichia coli CusB adapter protein

PLOS ONE

Dear Prof. Ruthstein, dear Sharon

Thank you for submitting your manuscript to PLOS ONE. After careful consideration, we feel that it has merit but does not fully meet PLOS ONE’s publication criteria as it currently stands. Therefore, we invite you to submit a revised version of the manuscript that addresses the points raised during the review process.

As you will see, the three expert reviewers in general deemed your study very interesting and worthy of publication. Especially reviewer 2 raised some issues concerning the data interpretation that I would ask you to take into account in particular.

We would appreciate receiving your revised manuscript by Sep 21 2019 11:59PM. To enhance the reproducibility of your results, we recommend that if applicable you deposit your laboratory protocols in protocols.io, where a protocol can be assigned its own identifier (DOI) such that it can be cited independently in the future. For instructions see: http://journals.plos.org/plosone/s/submission-guidelines#loc-laboratory-protocols

We look forward to receiving your revised manuscript.

Kind regards,

Dariush Hinderberger

Academic Editor

PLOS ONE

Journal Requirements:

Reviewers' comments:

Reviewer's Responses to Questions

**Comments to the Author**

1. Is the manuscript technically sound, and do the data support the conclusions?

Reviewer #1: Yes

Reviewer #2: Partly

Reviewer #3: Yes

2. Has the statistical analysis been performed appropriately and rigorously? 

Reviewer #1: Yes

Reviewer #2: N/A

Reviewer #3: N/A

3. Have the authors made all data underlying the findings in their manuscript fully available?

Reviewer #1: Yes

Reviewer #2: Yes

Reviewer #3: Yes

4. Is the manuscript presented in an intelligible fashion and written in standard English?

Reviewer #1: Yes

Reviewer #2: No

Reviewer #3: Yes

5. Review Comments to the Author

Reviewer #1: Developing novel drugs to eliminate new pathogenic bacteria is one of the highest priority in antibiotics research. One of the best approaches is to inhibit copper transporters in prokaryotic systems which is required for many crucial biological pathways. Pathogenic systems contain a highly sophisticated copper-regulation network. Recently, it was revealed that the CusB protein part of the CusCBA periplasmic Cu(I) efflux system in Gram-negative bacteria, play an important key role in the functioning of the whole CusCBA system

In this manuscript, the authors targeted two methionine residues (M227 and M241) that are essential for the proper function of CusB of Escherrichia coli. The authors used a combination of in-vitro spectroscopic measurements and cell experiments to study the significance of each single methionine residue in CusB and its effect on cell survival. Indeed, they used pulsed EPR spectroscopy to study the conformational changes of CusB associated with Cu(I) binding.

The combination of the in-vitro structural measurements comprised of double electron-electron resonance (DEER) spectroscopy, coupled with site directed spin labeling (SDSL). Also, the DEER technique pulsed electron double resonance (PELDOR) was used to measure the dipolar interactions between two or more electron spins. The authors compared the DEER data with the structure of CusB and used multiscale modeling of macromolecular systems.

Moreover, the authors evaluated the binding affinity of CusB proteins for Cu(I) by using the spectrophotometric competition assay with bicinchoninic acid (BCA) as a ligand. Furthermore, the authors conducted live/dead fluorescence cell imaging experiments to determine whether point mutations might lead to cell death or only to a reduction in E. coli cell growth.

In this excellent rationale, they showed that beside essential methionine residues reported before, there were two additional methionine residues at CusB domains 2 and 3 (M227 and M241) that are important for proper function of the CusCBA transporter. Mutating these methionine residues inhibits conformational changes in CusB in the presence of Cu(I) and results in a non-functional CusB protein, decreased copper resistance, and increased cell toxicity in pathogenic cells.

This manuscript is very well written and it was a pleasure reading it. The biochemical analytical work and associated single-point mutation genetics is outstanding.

Tow minor comments are indicated:

1. There are lots of scientific abbreviation like: multicopper oxidase CueO, CopA, CusCBA that should be defined for the non- biochemistry expert.

2. In page ,lines 71 and 72 you have written the following;

The inner and the outer membrane proteins are connected by a linker protein, CusB, in the periplasm, and are at a CusA: CusB: CusC oligomerization ratio of 3:6:3 [16].

QUERY: Do you mean that the inner and the outer membrane proteins are connected by a linker protein, CusB, in the periplasm, which is composed of a CusA: CusB: CusC oligomerization ratio of 3:6:3 [16].

If that’s what you mean, than my correction seems better

Reviewer #2: The study by Meir et al. described in this manuscript focuses on the effects of Cu(I) binding to CusB, being part of the periplasmic CusCBA copper-I efflux system of (pathogenic) gram-negative bacteria. The authors combined structural investigations using EPR (DEER) spectroscopy, photometric Cu(I)-binding assays and live cell experiments to obtain insights into the functional relevance of several methionine residues in CusB.

The subject of this study is of high relevance for the field of antibiotics research, opening pathways to new antibiotics specifically targeting the Cus system of pathogenic bacteria that is not present in mammalian cells.

The concept of this study is well elaborated, the experiments appear to be performed expertly from a technical point of view, and the results obtained in this study reveal valuable insights into the function of the CusB protein, esp. its Cu(I) binding properties.

Nevertheless, interpretation of the data and its representation in the form of the present manuscript do not appear appropriate for publication without major revisions as detailed in the following.

1. Major points to address:

- The authors state that expression levels (Fig. S5) and secondary structure (Fig. S4, stated in l. 127/8 and l. 156) of all mutant clones are identical. Apart from the fact that this is only shown for 4 of the 10 mutants prepared, the CD spectra depicted in Fig. S4 do show clear differences between wt and the mutants. These differences are obvious and if it is stated that the secondary structure is not affected by the met mutations, this should be clearly shown by quantitative analyses of the CD spectra in terms of secondary structure content.

- Analysis and interpretation of the DEER data needs to be elaborated in more detail. The statement "..., whereas in the holo-state the spin labels are found in a symmetrical orientation, which results in a much narrower distance distribution" is not clear. What do you mean by "symmetrical orientation" of the labels and what does that mean for the overall structure of the protein (dimer).

I also do not see, that the structural changes are "inhibited" in the mutants 227, 241 and 64 to various extents, as I cannot clearly identify a simple shift in an equilibrium between two populations in the distance distributions. Especially when comparing the distance distributions for CusB_M241I, M227I and M64I with holo- and apo-CusB, it appears that the structural alterations upon Cu(I) binding are more complex for the mutants than a simple two-state equilibrium.

Another minor point concerning the comparison with the RLA results: It has been shown that the RLA for MTSSL calculated at 298 K performs better than the one for 175 K.

- Refering to the structural models shown in Fig. 6 with the positions of M227 and M241, the authors predict two Cu(I) binding sites in the CusB dimer comprised by these residues, respectively. Nevertheless, as stated in the text, these residues exhibit Calpha-Calpha distances > 2 nm, making it very unlikely that e.g. the two M241 residues in the dimer (2.05 nm in the holostate) can together coordinate one Cu(I) ion. Thus I do not see any indications for such Cu binding site based on the results shown here.

Also, the prediction for another binding site comprised by M227 residues from interacting dimers in the CusCBA complex is not supported by any structural model or experimental data obtained with the assembled complex. The major question concerning this predicted binding site, that is if the CusB dimers could assemble in an appropriate orientation for bringing two M227 residues in close proximity, could be addressed for example with docking approaches.

- In the discussion section the authors state that mutation of M227 or M241 lead to "nonfunctional" CusB, what is simply not true. Both mutants still bind Cu(I) (cf. Table 1), and the affinities are only reduced ~2-fold. Thus I would describe it as reduced or impaired functionality, what is also reflected in the clear but not dramatically decreased growth rates observed in the cell experiments.

- English writing is poor in some parts of the manuscript, often leading to sentences that are difficult to understand (see below for the most relevant examples). I recommend to let a native speaker revise the manuscript.

2. Formal (textual) points to address:

- Fig. 1: the domain structure of CusB should be indicated (e.g. by different colors for the 4 domains) in the structure in panel B. It would also be very helpful to indicate the positions of the Met residues (esp. of the four mutants addressed in detail in this manuscript) in this figure. The same for Fig. 3C.

- Fig. 2: Panels A and B are "horrible". Esp. in panel A one cannot identify the single growth curves. I would suggest to show only the growth curves of the four "relevant" mutants in this figure and shift the other curves to the SI. Furthermore, in the main text only the final OD's after 14 h of growth are discussed (panel C) and not the growth kinetics. Thus, Fig. 2 could be reduced to show only panel C and all growth curves could be placed in the SI. Alternatively, the growth kinetics need to be (at least briefly) described in the text.

- l. 70/71: I guess you mean "...and an outer membrane protein (CusC)".

- l. 125: "..., we introduced a single point mutation in one of them." - not clear, please reformulate

- l. 136/7: " ...under Cu(II) was reduced by..., compared with wild-type (WT) CusB after 14 h..." - I guess you mean that the growth rate of the WT was reduced under Cu(II) stress compared to Cu-free growth conditions.

- l. 181: I do not understand what you mean with "...the distance distribution between four spin labels was consumed"!

- ll. 183-190: The description of the rotamer library approach implemented in MMM and used here is not well described. First, the authors did not perform multiscale modeling, they only used the RLA apprach implemented in the software. Second, MMM is not a computational approach that was used to derive the rotamer library. The RLA is only implemented in MMM.

- l. 248: An increased kD value does not weaken the binding, the weaker binding is reflected by the increased kD.

- l. 274: What is "An identical imaging comparison..." ?

- l. 305/6: I do not understand what "...it indicates that within 8 hr of selective CusB Met residue inhibition" means.

- l. 311: omit "numerically"

- SI, tables S1, S2, S3: Please reduce the number of decimal digits (Max. 1-2) for columns [Cu(I)-BCA2] and [Cu(I)-protein].

Reviewer #3: This study seeks to extend the understanding of the role of the various methionine residues present in copper efflux system (CusCBA). In addition to the important residues identified in previous studies the authors have identified two additional residues of importance for the function of the CusB protein. In particular, mutations of these residues led the authors to identifying an important inhibition of a conformational change upon Cu binding.

This is a very nice multi-technique study that uses a combination of cell growth studies, DEER (to probe conformational changes), and spectrophotometric assays to determine Cu(I) binding. The entire package presents a very convincing argument and strongly suggests further study as to the detailed role of the residues identified. I have some minor comments below but the manuscript is fit for publication.

Comments

Cell growth studies

I do not feel especially qualified to comment on the cell growth studies, however they appear to be well done and support the conclusions drawn in the paper.

I have one question in re: to the comment: “…slight effect caused by the minimal copper concentration naturally present…and the significance of the methionine residues…” Is it known how much copper is present in the growth medium?

DEER studies

This is an ideal use of DEER, and is well done. Some comments:

Line 181: “…the distance distribution was consumed…” I am not sure what this sentence is saying. Do the authors mean conserved?

Figure 3B caption: missing space after “…CusB+Cu(I)”. Should read “…+Cu(I) structure models.”

Cu(I) affinity studies by UV-Vis

Some concerns with the data:

Figure 4A: While the peak height is decreasing there appears to be a baseline shift as well. Figure 4B shows clean isobestic points, but these appear absent in 4A. Could the authors comment about this? The derived Kd values appear reasonable, however.

Cell viability studies

Similar to above, I am not qualified to comment in a significant way, but these appear to be well done and support the overall conclusions.

Would it be possible to merge these studies with the cell growth studies? May shorten the manuscript in a useful way. They seem highly related (though this very well may just be my lack of experience with these types of microbiological studies).

General comments

This is a well-written and concise manuscript that forms a convincing argument as to the newly identified importance of the two residues of interest. The data is high quality and a significant amount of important supplemental data has been included.

6. PLOS authors have the option to publish the peer review history of their article (what does this mean?). If published, this will include your full peer review and any attached files.

Reviewer #1: Yes: Joseph Banoub

Reviewer #2: No

Reviewer #3: No

---

## [Author Response · Author response to Decision Letter 0]

12 Aug 2019

Reviewer #1: Developing novel drugs to eliminate new pathogenic bacteria is one of the highest priority in antibiotics research. One of the best approaches is to inhibit copper transporters in prokaryotic systems which is required for many crucial biological pathways. Pathogenic systems contain a highly sophisticated copper-regulation network. Recently, it was revealed that the CusB protein part of the CusCBA periplasmic Cu(I) efflux system in Gram-negative bacteria, play an important key role in the functioning of the whole CusCBA system

In this manuscript, the authors targeted two methionine residues (M227 and M241) that are essential for the proper function of CusB of Escherrichia coli. The authors used a combination of in-vitro spectroscopic measurements and cell experiments to study the significance of each single methionine residue in CusB and its effect on cell survival. Indeed, they used pulsed EPR spectroscopy to study the conformational changes of CusB associated with Cu(I) binding.

The combination of the in-vitro structural measurements comprised of double electron-electron resonance (DEER) spectroscopy, coupled with site directed spin labeling (SDSL). Also, the DEER technique pulsed electron double resonance (PELDOR) was used to measure the dipolar interactions between two or more electron spins. The authors compared the DEER data with the structure of CusB and used multiscale modeling of macromolecular systems.

Moreover, the authors evaluated the binding affinity of CusB proteins for Cu(I) by using the spectrophotometric competition assay with bicinchoninic acid (BCA) as a ligand. Furthermore, the authors conducted live/dead fluorescence cell imaging experiments to determine whether point mutations might lead to cell death or only to a reduction in E. coli cell growth.

In this excellent rationale, they showed that beside essential methionine residues reported before, there were two additional methionine residues at CusB domains 2 and 3 (M227 and M241) that are important for proper function of the CusCBA transporter. Mutating these methionine residues inhibits conformational changes in CusB in the presence of Cu(I) and results in a non-functional CusB protein, decreased copper resistance, and increased cell toxicity in pathogenic cells.

This manuscript is very well written and it was a pleasure reading it. The biochemical analytical work and associated single-point mutation genetics is outstanding.

Author’s reply: thank you very much.

Two minor comments are indicated:

1. There are lots of scientific abbreviation like: multicopper oxidase CueO, CopA, CusCBA that should be defined for the non- biochemistry expert.

Author’s reply: Abbreviation table was added to the manuscript

2. In page ,lines 71 and 72 you have written the following;

The inner and the outer membrane proteins are connected by a linker protein, CusB, in the periplasm, and are at a CusA: CusB: CusC oligomerization ratio of 3:6:3 [16].

QUERY: Do you mean that the inner and the outer membrane proteins are connected by a linker protein, CusB, in the periplasm, which is composed of a CusA: CusB: CusC oligomerization ratio of 3:6:3 [16].

If that’s what you mean, than my correction seems better

Author’s reply: thanks - we corrected the sentence according to your suggestion.

Reviewer #2: The study by Meir et al. described in this manuscript focuses on the effects of Cu(I) binding to CusB, being part of the periplasmic CusCBA copper-I efflux system of (pathogenic) gram-negative bacteria. The authors combined structural investigations using EPR (DEER) spectroscopy, photometric Cu(I)-binding assays and live cell experiments to obtain insights into the functional relevance of several methionine residues in CusB.

The subject of this study is of high relevance for the field of antibiotics research, opening pathways to new antibiotics specifically targeting the Cus system of pathogenic bacteria that is not present in mammalian cells.

The concept of this study is well elaborated, the experiments appear to be performed expertly from a technical point of view, and the results obtained in this study reveal valuable insights into the function of the CusB protein, esp. its Cu(I) binding properties.

Nevertheless, interpretation of the data and its representation in the form of the present manuscript do not appear appropriate for publication without major revisions as detailed in the following.

Author’s reply: Thank you for your comments – they are very helpful, we addressed them below based on your suggestions.

1. Major points to address:

- The authors state that expression levels (Fig. S5) and secondary structure (Fig. S4, stated in l. 127/8 and l. 156) of all mutant clones are identical. Apart from the fact that this is only shown for 4 of the 10 mutants prepared, the CD spectra depicted in Fig. S4 do show clear differences between wt and the mutants. These differences are obvious and if it is stated that the secondary structure is not affected by the met mutations, this should be clearly shown by quantitative analyses of the CD spectra in terms of secondary structure content.

Author’s reply: The quantitative analysis of the CD spectra was added to the SI (Figure S5). Ten mutations were prepared for the cell experiments, in vitro experiments were performed on four mutations, and therefore for the CD we used them. The CD suggests less than 3% effect on the secondary structure. We revised the conclusion in the SI, and deleted the above statement from the text.

- Analysis and interpretation of the DEER data needs to be elaborated in more detail. The statement "..., whereas in the holo-state the spin labels are found in a symmetrical orientation, which results in a much narrower distance distribution" is not clear. What do you mean by "symmetrical orientation" of the labels and what does that mean for the overall structure of the protein (dimer).

I also do not see, that the structural changes are "inhibited" in the mutants 227, 241 and 64 to various extents, as I cannot clearly identify a simple shift in an equilibrium between two populations in the distance distributions. Especially when comparing the distance distributions for CusB_M241I, M227I and M64I with holo- and apo-CusB, it appears that the structural alterations upon Cu(I) binding are more complex for the mutants than a simple two-state equilibrium.

Another minor point concerning the comparison with the RLA results: It has been shown that the RLA for MTSSL calculated at 298 K performs better than the one for 175 K.

Author’s reply: We rewrote the DEER section, Figure S2 was added to the SI, and we hope that with these changes it is more clear. The DEER section was written as follow: “To compare the DEER data with the structure of CusB, we used the MMM software (2015 version) [39]. The software requires an entry PDB file, and the residues where the spin-labels are attached to the protein. This method describes spin labels by a set of alternative conformations or rotamers, which can be attached without serious clashes with atoms of other residues or cofactors. The rotamer library is derived from molecular dynamic simulation with a total length of 100 ns at a temperature of 175K, which is an estimate of the glass transition of a protein sample. Here we used the two structures of apo-CusB and holo-CusB, which were previously constructed based on various DEER constraints (Figure 1B) as an input PDB files for the MMM software [22]. The dashed black-line distribution in Figure 3B denotes the distance distribution obtained from MMM. The apo-state exhibits a broad distance distribution that is consistent with the calculated model structure of CusB, whereas in the holo-state the structure is more compact and rigid, which results in a much narrower distance distribution [22]. The orientation of the spin-labels in the two structures are presented in Figure S2, SI. DEER also confirmed that when one specific methionine residue among M64, M227, and M241 was mutated, structural changes in CusB protein associated with Cu(I) binding were affected. For simplicity, we used a linear composition of the apo-WT-CusB distance distribution with the holo-WT-CusB distance distribution (see Figure S2, SI). Although the effect of each mutation on conformational changes may not be a simple two-step model between the apo- and holo-states, this approach can still provide us with an overview on how much each mutation disturbs the conformation and function of the CusB protein. In the case of CusB_M64I + Cu(I), the distance distribution is very similar to the apo WT-CusB distance distribution, suggesting that Cu(I) did not affect the conformation of CusB in the presence of M64I mutation. In contrast, for holo CusB_M398I, the distribution is slightly broader than the holo WT-CusB. However, the best fit was still obtained with 100% holo WT-CusB distance distribution, indicating that M398I did not affect the conformational changes of CusB upon Cu(I) binding. For CusB_M227I and CusB_M241I, the best fit was obtained with convolutions of 50% and 90%, respectively, of the apo-WT-CusB distance distribution with the holo-WT-CusB distance distribution. This suggests that both mutations affect the conformational changes of CusB in the presence of Cu(I), while M241I has a larger effect on conformation of CusB than the M227I mutation.”

We calculated the RLA both at 175K as well as 298K, there were no large differences. The 175K is a bit better fit for our data.

- Refering to the structural models shown in Fig. 6 with the positions of M227 and M241, the authors predict two Cu(I) binding sites in the CusB dimer comprised by these residues, respectively. Nevertheless, as stated in the text, these residues exhibit Calpha-Calpha distances > 2 nm, making it very unlikely that e.g. the two M241 residues in the dimer (2.05 nm in the holostate) can together coordinate one Cu(I) ion. Thus I do not see any indications for such Cu binding site based on the results shown here.

Also, the prediction for another binding site comprised by M227 residues from interacting dimers in the CusCBA complex is not supported by any structural model or experimental data obtained with the assembled complex. The major question concerning this predicted binding site, that is if the CusB dimers could assemble in an appropriate orientation for bringing two M227 residues in close proximity, could be addressed for example with docking approaches.

Author’s reply: We agree with the reviewer that based on the results presented in this manuscript, we cannot offer the structural model presented in Figure 6, therefore we deleted this part from the manuscript. However, we do think that the results clearly support the conclusion that M227 and M241 residues are important for proper function of CusB, which is the main conclusion of this study. 

- In the discussion section the authors state that mutation of M227 or M241 lead to "nonfunctional" CusB, what is simply not true. Both mutants still bind Cu(I) (cf. Table 1), and the affinities are only reduced ~2-fold. Thus I would describe it as reduced or impaired functionality, what is also reflected in the clear but not dramatically decreased growth rates observed in the cell experiments.

Author’s reply: Yes we agree, and it was corrected in the text.

- English writing is poor in some parts of the manuscript, often leading to sentences that are difficult to understand (see below for the most relevant examples). I recommend to let a native speaker revise the manuscript.

Author’s reply: A native speaker checked and revised the manuscript

2. Formal (textual) points to address:

- Fig. 1: the domain structure of CusB should be indicated (e.g. by different colors for the 4 domains) in the structure in panel B. It would also be very helpful to indicate the positions of the Met residues (esp. of the four mutants addressed in detail in this manuscript) in this figure. The same for Fig. 3C.

Author’s reply: As suggested, we revised Figure 1, and added the domains to Figure 3C.

- Fig. 2: Panels A and B are "horrible". Esp. in panel A one cannot identify the single growth curves. I would suggest to show only the growth curves of the four "relevant" mutants in this figure and shift the other curves to the SI. Furthermore, in the main text only the final OD's after 14 h of growth are discussed (panel C) and not the growth kinetics. Thus, Fig. 2 could be reduced to show only panel C and all growth curves could be placed in the SI. Alternatively, the growth kinetics need to be (at least briefly) described in the text.

Author’s reply: We moved Figure 2A and 2B to the SI as suggested by the referee.

- l. 70/71: I guess you mean "...and an outer membrane protein (CusC)".

Author’s reply: Corrected.

- l. 125: "..., we introduced a single point mutation in one of them." - not clear, please reformulate

Author’s reply: We rephrased it: “Ten Met residues exist in CusB; in order to determine which of them are essential for cell growth, each time one methionine residue was mutated to isoleucine.”

- l. 136/7: " ...under Cu(II) was reduced by..., compared with wild-type (WT) CusB after 14 h..." - I guess you mean that the growth rate of the WT was reduced under Cu(II) stress compared to Cu-free growth conditions.

Author’s reply: yes, we corrected it. 

- l. 181: I do not understand what you mean with "...the distance distribution between four spin labels was consumed"!

Author’s reply: The DEER section was rewritten – see the answer above.

- ll. 183-190: The description of the rotamer library approach implemented in MMM and used here is not well described. First, the authors did not perform multiscale modeling, they only used the RLA apprach implemented in the software. Second, MMM is not a computational approach that was used to derive the rotamer library. The RLA is only implemented in MMM.

Author’s reply: The DEER section was rewritten – see the answer above.

- l. 248: An increased kD value does not weaken the binding, the weaker binding is reflected by the increased kD.

Author’s reply: Corrected

- l. 274: What is "An identical imaging comparison..." ?

Author’s reply: thanks - we deleted it (it was out of context).

- l. 305/6: I do not understand what "...it indicates that within 8 hr of selective CusB Met residue inhibition" means.

Author’s reply: We rephrased it: “This finding indicates that introducing mutations in CusB results in toxicity that eventually leads to cell death within 8 hr.”

- l. 311: omit "numerically"

Author’s replay: Corrected.

- SI, tables S1, S2, S3: Please reduce the number of decimal digits (Max. 1-2) for columns [Cu(I)-BCA2] and [Cu(I)-protein].

Author’s reply: Was revised as suggested by the referee.

Reviewer #3: This study seeks to extend the understanding of the role of the various methionine residues present in copper efflux system (CusCBA). In addition to the important residues identified in previous studies the authors have identified two additional residues of importance for the function of the CusB protein. In particular, mutations of these residues led the authors to identifying an important inhibition of a conformational change upon Cu binding.

This is a very nice multi-technique study that uses a combination of cell growth studies, DEER (to probe conformational changes), and spectrophotometric assays to determine Cu(I) binding. The entire package presents a very convincing argument and strongly suggests further study as to the detailed role of the residues identified. I have some minor comments below but the manuscript is fit for publication.

Author’s reply: Thank you very much.

Comments

Cell growth studies

I do not feel especially qualified to comment on the cell growth studies, however they appear to be well done and support the conclusions drawn in the paper.

I have one question in re: to the comment: “…slight effect caused by the minimal copper concentration naturally present…and the significance of the methionine residues…” Is it known how much copper is present in the growth medium?

Author’s reply: Thanks - we added it to the text – it is 0.5 �M.

DEER studies

This is an ideal use of DEER, and is well done. Some comments:

Line 181: “…the distance distribution was consumed…” I am not sure what this sentence is saying. Do the authors mean conserved?

Author’s reply: We rephrased it in the text.

Figure 3B caption: missing space after “…CusB+Cu(I)”. Should read “…+Cu(I) structure models.”

Author’s reply: thanks – we corrected it.

Cu(I) affinity studies by UV-Vis

Some concerns with the data:

Figure 4A: While the peak height is decreasing there appears to be a baseline shift as well. Figure 4B shows clean isobestic points, but these appear absent in 4A. Could the authors comment about this? The derived Kd values appear reasonable, however.

Author’s reply: The spectra were baselined according to the absorption value at 800 nm, which is zero – we added it to the SI.

Cell viability studies

Similar to above, I am not qualified to comment in a significant way, but these appear to be well done and support the overall conclusions.

Would it be possible to merge these studies with the cell growth studies? May shorten the manuscript in a useful way. They seem highly related (though this very well may just be my lack of experience with these types of microbiological studies).

Author’s reply: The cell viability tests are complimentary to the cell growth experiments since they show that point mutations do not only affect cell growth but also are toxic to the cell. Therefore, we decided to leave this section to the end, since this is very important conclusion towards development of new antibiotics.

---

## [Decision Letter · Decision Letter 1]

16 Aug 2019

Exploring the role of the various methionine residues in the Escherichia coli CusB adapter protein

PONE-D-19-17365R1

Dear Dr. Ruthstein, dear Sharon,

I am pleased to inform you that your manuscript has been judged scientifically suitable for publication and will be formally accepted for publication once it complies with all outstanding technical requirements.

With kind regards,

Dariush Hinderberger

Academic Editor

PLOS ONE

Additional Editor Comments (optional):

Reviewers' comments:

Reviewer's Responses to Questions

**Comments to the Author**

1. If the authors have adequately addressed your comments raised in a previous round of review and you feel that this manuscript is now acceptable for publication, you may indicate that here to bypass the “Comments to the Author” section, enter your conflict of interest statement in the “Confidential to Editor” section, and submit your "Accept" recommendation.

Reviewer #2: All comments have been addressed

2. Is the manuscript technically sound, and do the data support the conclusions?

Reviewer #2: Yes

3. Has the statistical analysis been performed appropriately and rigorously? 

Reviewer #2: N/A

4. Have the authors made all data underlying the findings in their manuscript fully available?

Reviewer #2: Yes

5. Is the manuscript presented in an intelligible fashion and written in standard English?

Reviewer #2: Yes

6. Review Comments to the Author

Reviewer #2: (No Response)

7. PLOS authors have the option to publish the peer review history of their article (what does this mean?). If published, this will include your full peer review and any attached files.

Reviewer #2: No

---

## [Editor Report · Acceptance letter]

20 Aug 2019

PONE-D-19-17365R1 

Exploring the role of the various methionine residues in the Escherichia coli CusB adapter protein 

Dear Dr. Ruthstein:

I am pleased to inform you that your manuscript has been deemed suitable for publication in PLOS ONE. Congratulations! Your manuscript is now with our production department. 

With kind regards,

on behalf of

Professor Dr. Dariush Hinderberger 

Academic Editor

PLOS ONE